# Molecular Perspective of Nanoparticle Mediated Therapeutic Targeting in Breast Cancer: An Odyssey of Endoplasmic Reticulum Unfolded Protein Response (UPR^ER^) and Beyond

**DOI:** 10.3390/biomedicines9060635

**Published:** 2021-06-02

**Authors:** Safikur Rahman, Vijay Kumar, Anuj Kumar, Tasduq S. Abdullah, Irfan A. Rather, Arif Tasleem Jan

**Affiliations:** 1Department of Botany, Munshi Singh College, BR Ambedkar Bihar University, Muzaffarpur 845401, India; shafique2@gmail.com; 2Department of Biotechnology, Yeungnam University, Gyeongsan 38541, Korea; vijaykumarcbt@ynu.ac.kr; 3School of Chemical Engineering, Yeungnam University, Gyeongsan 38541, Korea; anuj.budhera@gmail.com; 4Council of Scientific and Industrial Research–Indian Institute of Integrative Medicine (CSIR–IIIM), Jammu 180001, India; stabdullah@iiim.res.in; 5Department of Biological Sciences, Faculty of Science, King Abdulaziz University (KAU), P.O. Box 80141, Jeddah 21589, Saudi Arabia; 6School of Biosciences and Biotechnology, Baba Ghulam Shah Badshah University, Rajouri 185234, India

**Keywords:** breast cancer, ER stress, nanotechnology, nanomedicine, therapeutics

## Abstract

Breast cancer (BC) is the second most frequent cause of death among women. Representing a complex and heterogeneous type of cancer, its occurrence is attributed by both genetic (gene mutations, e.g., BRCA1, BRCA2) and non-genetic (race, ethnicity, etc.) risk factors. The effectiveness of available treatment regimens (small molecules, cytotoxic agents, and inhibitors) decreased due to their poor penetration across biological barriers, limited targeting, and rapid body clearance along with their effect on normal resident cells of bone marrow, gastrointestinal tract, and hair follicles. This significantly reduced their clinical outcomes, which led to an unprecedented increase in the number of cases worldwide. Nanomedicine, a nano-formulation of therapeutics, emerged as a versatile delivering module for employment in achieving the effective and target specific delivery of pharmaceutical payloads. Adoption of nanotechnological approaches in delivering therapeutic molecules to target cells ensures not only reduced immune response and toxicity, but increases the stability of therapeutic entities in the systemic circulation that averts their degradation and as such increased extravasations and accumulation via enhanced permeation and the retention (EPR) effect in target tissues. Additionally, nanoparticle (NP)-induced ER stress, which enhances apoptosis and autophagy, has been utilized as a combative strategy in the treatment of cancerous cells. As nanoparticles-based avenues have been capitalized to achieve better efficacy of the new genera of therapeutics with enhanced specificity and safety, the present study is aimed at providing the fundamentals of BC, nanotechnological modules (organic, inorganic, and hybrid) employed in delivering different therapeutic molecules, and mechanistic insights of nano-ER stress induced apoptosis and autophagy with a perspective of exploring this avenue for use in the nano-toxicological studies. Furthermore, the current scenario of USA FDA approved nano-formulations and the future perspective of nanotechnological based interventions to overcome the existing challenges are also discussed.

## 1. Introduction

Cancer—a life threatening disease, is a major cause of global mortality and morbidity [1,2]. Irrespective of the slow momentum in the occurrence of some cancers, cancer (in particular lung, colorectal, breast, and prostate), is still leading and as such represents a major public health concern worldwide [3]. The world health organization in its report has estimated an unprecedented increase in the number of cases to 19.3 million by 2025 [4]. As estimated by the American Cancer Society, around 1.8 million (0.2 million for breast cancer) new cancer cases and 0.6 million (0.04 million for breast cancer) cancer related deaths are projected to occur in the US alone in 2020 [3]. For cancers, various treatment options are available under clinical settings, including conventional (chemotherapy, radiotherapy, and surgery) and novel modalities such as immunotherapy, phototherapy, and gene therapy. Chemotherapy has its own limitations; the prominent limitations being the non-specific distribution of chemotherapeutics in the body, the compromised effect to chemotherapeutic doses after repeated administration, and in being ineffective in the cessation of tumor growth, metastasis, and its recurrence. Under such circumstances, it becomes imperative to develop effective means of drug delivery systems that could enhance the therapeutic efficiency of drugs (chemotherapeutics singly or in combination with other treatment regimens). With the advancement in nanotechnological approaches, it has now become possible to integrate a large number of components with various customized targeting strategies, therapeutic agents, and controlled-release mechanisms within an architecture framed at nano-scale [5]. The superiority of nanoparticles (NPs) in overcoming the multi-layered stromal-cell barriers for deeper tumor infiltration and drug perfusion makes it a promising future strategy for clinical applications either as an anticancer agent carrier or contrast agents in the biomedical imaging [6,7,8]. The current review symbolizes information about types and the occurrence of breast cancer (BC), a shift in the treatment paradigm from conventional to nanotechnological approaches for target specific delivery of therapeutic entities, along with mechanistic insights on the nanoparticle-induced endoplasmic reticulum unfolded protein response (UPR^ER^) to overcome the resistance, and as such the progression of cancer cells towards gaining an insight into the development of novel therapeutics for employment in the treatment of breast cancer.

## 2. Methodology

The article covers the literature available from 1991 to 2021. The information was located, selected, and extracted preferably from scientific journals, books, and reports via, library and electronic search (Pubmed). Documentation of the available information from the literature helped in drafting the different sections of the manuscript.

## 3. Breast Cancer

Breast cancer (BC) is a disease characterized by uncontrolled cell division that changes the topology of the breast tissue; thereby resulting in a lump or mass that arises particularly in the lobules (milk glands; lobular carcinoma) or in the ducts (ductal carcinoma) connecting the lobule to the nipple. There are two categorization stages in BC: (1) the anatomic stage (as per AJCC; American Joint Committee on Cancer) representing the extent of BC; restricted to regional (spread to surrounding tissue or nearby lymph nodes) or exhibiting distant spread (spread to different organs and/or lymph nodes), and (2) the prognostic stage (as per SEER; Surveillance, Epidemiology and End Results) with reference to the display of estrogen receptor (ER), progesterone receptor (PR), levels of human epidermal growth factor receptor 2 (HER2; a growth promoting protein), and grade (resemblance of appearance of cancer under microscope to normal breast tissue) [3,9]. Based on the ER, PR, and HER2 expression profile, BC has been further divided into five intrinsic subtypes: luminal A, luminal B, HER2-enriched, basal, and claudin-low [10,11].

BC is the second leading cause of cancer related deaths among women worldwide. Although age and hereditary factors contribute as the predisposing factors to BC, women exhibiting an increased risk (>100-fold) are more prone to breast cancer as compared to men [3,12]. Representing a complex and heterogenous type of cancer, BC is characterized by multiple genetic alterations (the prominent being the inherited alterations in the BRCA1 and BRCA2 genes, which accounts for 5 to 10% of female BC and 15 to 20% of familial BC), which are used as its diagnostic and prognostic markers [13,14]. Compared to women exhibiting about 10% risk of developing BC in general population, women with pathogenic variants of BRCA1 and BRCA2 were found exhibiting around a 70% risk of developing BC [15]. A risk factor of 35% is observed among the elderly women of 70+ age with the pathogenic variant of PALB2 (another gene complementing BRCA2 in its working) [16]. Mutations in other genes such as TP53, PTEN, CDH1, and STK11 are also found to be associated with an increased risk for the development of BC [17]. Triple negative BC (TNBC; characterized by absence of ER, PR, and HER2 on tumor cell membrane) owes 15 to 20% of invasive BC subtypes [18,19]. Additionally, non-genetic risk factors such as race, ethnicity, exposure to diethyl-stilbestrol, excessive alcohol consumption, birth control and contraceptive use, hormone replacement therapy after menopause, lack of physical activity, etc., also contribute significantly to the occurrence of the disease [3,20].

## 4. Nano-Based Therapeutics: A Paradigm Shift in the Treatment of BC

Treatment of BC involves the use of small molecules that serve the purpose of drugs such as anti-microtubule agents (Vincristine, Paclitaxel, etc.) and cytotoxic agents (Doxorubicin, etc.) along with other inhibitors that proved effective in reducing the occurrence of the disease ([21]; Table 1).

In the treatment regimes, the small molecules were found exerting their anti-cancerous effects on other cells residing in bone marrow, hair follicles, and the gastrointestinal (GI) tract as well; thereby compromising the function of the immune system, loss of hair, and inflammation of the GI tract [21]. Additionally, drugs exhibiting poor penetration across the biological barriers that control drug permeation, limited targeting and rapid clearance from the body, which limits the clinical outcome of the drugs [22,23]. With this, creating effective delivery modules that ensure efficient delivery of drugs to the target cells with reduced immune response and toxicity was explored, which ended with the adoption of nanotechnological approaches. These approaches ensure increased bioavailability and stability of the drugs in systemic circulation, which averts their degradation and activation of immune response along with efficient delivery to target the following increased extravasations and accumulation via, enhanced permeation and retention (EPR) effect [24,25,26,27,28,29]. Drug encapsulation and efficient delivery modules help in reducing the toxicity of drug cargoes to non-target organs.

Strategic improvement over the decades for long term success in the delivery of drugs has resulted in the development of three different nanotechnological platforms: (1) organic (polymers, liposomes, dendrimers, etc.), (2) inorganic (gold, iron oxide, silica, etc.), and hybrid (liposome filled with magnetic nanoparticles, metal-organic frameworks, etc). In the organic system, therapeutic cargo is either passively loaded or forcibly incorporated for its controlled release [30,31]. Hybrid nanotechnological platforms retain the stability and function of inorganic platforms and biocompatibility feature of the organic ones [32,33]. The information on size, synthesis, stability, and other functions of nanotechnological platforms are discussed in several excellent papers published over the years ([31], and references therein) and are hence not included here.

### 4.1. Polymeric-Based Nanoparticles

Organic polymers (50 nm to 10 µm) are categorized into; (1) natural polymers (alginate, chitosan, gelatin, etc.), (2) synthetic polymers (hydrogels, polyethylene glycol, polyethyleneimine, etc.), and (3) degradable polymers (collagen, polycaprolactone, etc.). The biocompatibility and reduced toxicity of the polymeric substances makes them a preferable choice for employment in the synthesis of nanoparticles [34,35,36]. Polymers of natural origin prepared by encapsulating therapeutic moieties without any modification in its makeup constitutes a preferable choice of delivering proteins, DNA, oligonucleotides, and drugs. In addition to basic methods of encapsulation such as emulsification, nanospray, and nanoprecipitation, the PRINT (particle replication in non-wetting template) method given by Xu et al. (2013) provides a new route of customizing encapsulation of different therapeutic moieties in uniformly sized polymeric nanoparticles [37]. Abraxane (a nanoparticle formulation of paclitaxel encapsulated in serum albumin) has been used in clinics for the treatment of metastatic BC [38]. Another nano-formulation of doxorubicin encapsulated in the magnetic nanoparticles with albumin coating conjugated to monoclonal antibodies to VEGF (vascular endothelial growth factor) increases the targeting of VEGF in the tumor [39]. Its magnetic core makes it possible for utilization in the diagnosis as it is detected by MRI on injection to tissues. On one side, where Poly-L-glutamic acid (PLGA; a synthetic biodegradable polymer) is employed for conjugate synthesis, polyethylene glycol (PEG) and N-(2-hydroxypropyl)-methacryl amide copolymer (HPMA) are the most widely used non-biodegradable polymers of synthetic origin [40,41,42,43]. PLGA (Poly-(lactic-co-glycolic acid) nanoencapsulated with *Callistemon citrinus* phenolics exhibited anticancer properties against BC cell lines viz. MDA-MB231, MCF-7 and MCF 10A [44]. PK1 (conjugate of doxorubicin with HPMA) is undergoing clinical trials for its possible employment in the treatment of cancer [45]. Folic acid conjugated β-lactoglobulin nanoparticles loaded with doxorubicin were found effective against MCF-7 and MDA-MB 231 [46]. Folic acid conjugated β-lactoglobulin nanoparticles prevent the release of doxorubicin into healthy cells as well as in the systemic circulation [46]. The release of doxorubicin occurs at the tumor site in response to an acidic environment; thereby presenting a higher toxic profile against BC cell lines even after 72 h compared with the free doxorubicin. Polymeric micelles-loaded drugs serve the purpose of delivering hydrophobic drugs by accommodating them in the hydrophobic core, while its hydrophilic shell renders it with the water solubility exploited for IV administration of the drug [47,48]. Genexol-PM (PEG-poly (D,L-lactide)-paclitaxel) is the first micelle formulation of drug paclitaxel [49]. Succinobucol with P188 (poloxamer) combination exhibiting 13-fold better bioavailability and the potential to inhibit vascular cell adhesion molecule -1 (VCAM-1), emerged as the best oral treatment for BC [50]. Polymeric nanoparticle PLGA-b-PEG based delivery of antisense-miR-21 and -miR10b and siRNA against multidrug resistant protein with doxorubicin cargo caused a significant reduction in the tumor growth and volume [51,52]. In triple negative BC (TNBC) models, RGD (Arg-Gly-Asp) ligand was found facilitating either target-specific drug delivery or inhibiting cancer invasion. RGD-conjugated polymer lipid nanoparticle with cargo of doxorubicin and mitomycin C (RGD-DMPLN) displayed enhanced cytotoxicity in the TNBC mouse models [53,54].

### 4.2. Liposomal Nanoparticles

Liposomes (400 nm) are lipid-based (phospholipids, triglycerides, and cholesterol) nanocarriers capable of carrying the therapeutic payload either embedded in the lipid bilayer or compartmentalized in the aqueous core. Liposome nanoparticles are prepared by reverse phase evaporation, solvent injection, and by extrusion methods [55]. Their biodegradable nature and compatibility with the biological environment and less toxicity compared with other delivery modules make them idealistic for delivery of the drug [56,57,58,59]. As tumor environments are acidic, the development of pH sensitive liposomes that remain stable at a physiological pH but collapse at acidic pH helps in the target-specific delivery of therapeutic payloads such as neoplastic drugs, recombinant proteins, etc. [60,61,62,63]. Liposome encapsulated doxorubicin/Paclitaxel exhibit remarkable enhancement in its anticancer activity compared to treatment with free drug molecule [64,65,66]. The development of liposomes with the capacity to accommodate two drug molecules simultaneously was found more effective in exerting anti-cancerous activity. Meng et al. (2016) observed that dual drug (reservatol and paclitaxel) developed liposomes were found effective in combating the resistance of cancer [67]. Adoption of the same strategy in encapsulating doxorubicin together with gadoteridol were used in developing long-circulating liposomes that release their payload in a target specific manner on applying ultrasound [68,69]. This strategy successfully achieved target-specific delivery of drugs with complete tumor regression in a breast tumor model. On testing the effect of nano-emulsion of doxorubicin with bromotetratrandrine (P-glycoprotein inhibitor) on MCF-7 cell line, increased doxorubicin uptake and accumulation with reduced cardiac toxicity was observed [70]. Doxorubicin loaded on to ER targeting long-circulating liposomes improved its cellular uptake and accumulation of drug in the tumor; thereby improving the drug release profile of doxorubicin in the treatment of BC [71]. Dai et al. (2014) exploited integrin α3 (a highly expressed receptor in TNBC cell line MDA MB-23) in treating TNBC with dual drug, doxorubicin, and rapamycin encapsulated in long-circulating liposomes with an attachment of cyclic octapeptide and observed enhanced antitumor activity compared to free drug [72]. Similar effects were observed for liposomes encapsulating a combination of doxorubicin and sorafenib [73]. Co-delivery of antagomir-10b and PTX were found reducing tumor growth and as such lung metastasis of BC [74]. A significant reduction (~80%) in the tumor growth was observed for PEG coated PTX-nanocrystal formulation in nude mice and lung tumor metastasis model [75]. Additionally, a reduction (~87%) in the tumor growth was observed for lipid conjugated estrogenic (bioactive form) nanoparticles having therapeutic load of cisplatin [22].

### 4.3. Dendrimer Nanoparticles

Dendrimers (10–100 nm) are synthetic macromolecules of multi-branched polymers prepared from the repeated monomer units following a convergent or divergent synthesis protocol, with a hydrophobic core that accommodates for therapeutic cargo surrounded by hydrophilic periphery [76,77]. The peripheral layer consists of multivalent functional groups that interact with the charged polar molecules, while its interior boundary encapsulates uncharged non-polar molecules through multiple interactions. Hydrophobic drugs (Paclitaxel, Doxorubicin) are frequently encapsulated in dendrimers through covalent interactions [78,79]. The outer functional layer controls the release of drug molecules in response to a change in the pH or after encountering certain specific enzymes. Wang et al. employed them in delivering antisense oligo (AODNs) conjugated to polyamidoamine dendrimers via, utilization of VEGF as AODNs receptor and observed a significant reduction in the tumor vascularization in TNBC xenograft mouse model [80]. Acting as vehicles for siRNA delivery, Finlay et al. used siRNA conjugated polyamidoamine dendrimer and observed a down regulation in the TWIST transcription factor in the TNBC model [81]. Linear dendrimeric copolymers blocks of polyamidoamine dendrimer and PEG with or without galactose and amphiphilic copolymer deblock of hydrophilic PEG and polymethacrylate hydrophobic block with acid labile side chains exhibited excellent biodistribution and high biocompatibility; and are thereby capable of achieving an enhanced permeation and retention (EPR) effect with greater efficiency to penetrate solid tumors for achieving their effect [82,83].

### 4.4. Inorganic Nanoparticles

Inorganic nanoparticles with optical, thermal, electrical, and magnetic properties at the expense of biocompatibility are preferred over organic form for desired clinical outcomes. Inorganic nanoparticles include a wide range of materials; (1) Metals such as gold, silver, zinc oxide, etc.; (2) Magnetic materials such as iron oxide, etc.; (3) Quantum dots; and (4) carbon-based materials. Metallic nanoparticles adopt diverse mechanisms in combating the growth of cancerous cells prominent in (1) the intracellular production of reactive oxygen species (ROS); (2) the enhancement in oxidative stress; (3) the progression of tumor cells to apoptotic pathway; and (4) the induction of hyperthermia (property of transition metals, i.e., conversion of electromagnetic radiation to heat) [84]. Of the metallic materials, gold, being relatively inert, is extensively used in the synthesis of versatile nano-vehicles such as nanorods (AuNR), nanoshells (AuNS) and nanocages (AuNC) with wide biological applications in the delivery of anticancer drugs such as paclitaxel [85,86,87,88,89]. The release of encapsulated therapeutic molecules such as proteins, DNA or RNA delivered as gold nanoparticles at q target site are made on the basis of their photo-physical properties. PEGylation of metallic nanoparticles has been found to increase their stability in the systemic circulation [90]. PEGylated gold nanoparticles together with ionizing radiation increase the survival in mouse models of breast cancer [91]. The use of gold nanoparticle based therapeutic modules with radiation therapy was found to be associated with the induction of radiosensitization, a phenomenon of increasing the susceptibility of tumor cells to radiation (discussed in detail in the studies of Turnbull et al. [92], Penninckx et al. [93], and Schuemann et al. [94]). The studies have reported that dose enhancement of the ionizing radiations could excite or ionize surrounding atoms, thereby leading to a series of ionizing events with a possible formation of the free radicals capable of damaging biological macromolecules, such as the double strand breaks (DSBs) in DNA [93,95,96,97]. Additionally, NPs induce alterations in the regulation of cellular genes including the genes of DNA damage repair such as thymidylate synthase (an important enzyme in the DNA damage repair), contribute to cell sensitization, which ultimately progresses with the killing of the cells [98,99,100]. Similarly, serum coated gold nanoparticles were found to inhibit migration and therefore the invasion of the cancer cells under both in vivo and in vitro conditions via down-regulation of the genes associated with energy generation. Cisplatin-loaded gold nanoparticles guided by laser leads to the inhibition of TNBC tumor and its metastasis to the nearby tissues [22].

Silver nanoparticles mediated enhancement of oxidative stress via an increase in ROS production proceeds with the cellular damage, which exerts pressure on the cell for progression to apoptosis [101,102]. These nanoparticles limit metastasis via inhibition of the VEGF. In gliomas, treatment with silver nanoparticles followed by radiotherapy proved effective in reducing the tumor [103]. Similar to silver nanoparticles, zinc oxide nanoparticles increase progression of the cell to apoptotic pathway [104]. With reduced cell toxicity, zinc oxide nanoparticles encapsulating cisplatin and paclitaxel increases the efficacy of these drugs in the BC cell line [105]. In another study, zinc oxide nanoparticles encapsulating paclitaxel and doxorubicin in conjugation with asparaginase were found effective in the target-specific delivery of these drugs [106]. Magnetic materials such as iron oxide (magnetite) exist as another variant that can be adopted in the synthesis of nanoparticles for the delivery of drugs in a target specific manner [107,108]. With less intrinsic magnetization unless applied with a magnetic field, ferromagnetic nanoparticles with therapeutic payload exhibiting less aggregation in colloidal suspension have been used in the treatment of cancer [109]. Quantum dots (semiconductor material with broad excitation spectrum) have also been administered as a delivery vehicle in the target specific delivery of therapeutic molecules [110,111,112]. Inorganic nanoparticles capable of responding to changes in pH have been worked to facilitate the enhancement in the stability of drugs in systemic circulation and their delivery to target tissues [113,114,115]. Zinc oxide quantum dots nanoparticles conjugated to folic acid with a doxorubicin payload remains stable at physiological pH [116]. With less toxicity, the release of the doxorubicin payload occurs in response to acidic condition at the tumor site, ensures stability and tumor-specific delivery of the drug. Carbon-based materials such as carbon nanotubes, etc., which employ its structural property in getting penetrated in the cell, have also been employed in achieving stability, prolonged distribution and target specific delivery of small molecules (antisense oligonucleotides, short interfering RNA) and drugs (doxorubicin, paclitaxel) [109]. A list of different types of nanoparticles inducing ER stress in breast cancer and other cancer cell line is given in Table 2.

### 4.5. Hybrid Nanoparticles

Hybrid platforms represent a combination of organic with the inorganic components that presents exciting properties compared with their individual counterparts [131]. As a multifunctional platform, it merges the biocompatibility of organic platforms with stability of structures that promotes target specific drug delivery property of the inorganic counterpart. Examples of hybrid nanotechnological platforms include metal organic frameworks [132,133], liposome based magnetic nanoparticles [134,135,136], and coordination polymer nanoparticles [137,138]. Graphene oxide hybrid with magnetic materials offer biocompatibility and controlled drug-release property. Doxorubicin loaded graphene oxide—Fe_3_O_4_ yolk shell nanoparticles exhibited high loading capacity, high biocompatibility, perfect dispersibility, and pH response drug release [139].

## 5. Mechanistic Insights of UPR Activation in Breast Cancer Treatment

The most prominent feature of a cancerous cell is uncontrolled cell division without obeying the normal signaling cascade. In general, cancer cells encounter increased cellular stresses, which include mitotic stress, oxidative stress, proteotoxic stress, etc. These stress pathways are the route for the survival of cancer cells. The dependence of cancer cells on these pathways for survival can be exploited in cancer therapy either as stress sensitization or stress overload [140]. Endoplasmic stress (ER) stress response is one such type of stress response in which ER launches various coping mechanisms to alleviate the damage, allowing the cell to adapt to such stresses. If the recovery of cellular adaptation is not achieved, it prolongs ER stress and thereby triggers apoptosis. Any drug targeting the ER response pathways can potentially act synergistically in inducing stress overload among cancer cells, leading to enhanced cancer cell killing or reduced side effects. Several tumor cells, including breast, gastric, and hepatocellular carcinoma cells showed altered unfolded protein response (UPR) [141,142,143]. Studies have suggested that the selective inhibitor of 26S proteasomal system, bortezomib (Velcade), approved for treating multiple myeloma and mantle cell lymphoma, results in the accumulation of ERAD (endoplasmic reticulum-associated degradation) causing ER stress, and contributes to its cytotoxic activity against cancer cells [144,145,146,147]. As cancer cells are sensitive to ER stress pathway, the selective activation of ER stress pathway by NPs exposure could serve a possible way in the treatment of different cancers.

### 5.1. ER Stress Response Pathway

In a cell, ER is an organelle of high functionality with respect to the folding of proteins into a stable and functionally correct conformation. Under normal conditions, a cell either restricts the misfolded proteins in the ER lumen, where molecular chaperons or enzymes assists them to get properly folded for retaining their functional state, or directs them to cytosol for degradation via a ubiquitin-proteosome system (UPS; ~80% cellular proteins follow the UPS path for degradation) or autophagy-lysosome pathway (primary degradation route for misfolded or aggregated proteins) [148]. Playing a crucial role in cellular homeostasis and cell survival, any impediment in the protein folding capacity of ER leads to enhancement in the accumulation of misfolded or aggregated proteins, a condition referred to as ER stress [149]. Characterized by a high level of misfolded or aggregated proteins, ER stress induction greatly influences the fate of cells, which ultimately progresses to the development of different diseases in humans [150,151]. Dysfunction in the functional state of ER triggers signaling across the cascade, termed as unfolded protein response (UPR^ER^) mediated by transmembrane stress sensors resident on the ER towards restoration of the ER homeostasis [149,152,153,154]. ER stress signal is sensed by three transmembrane resident stress sensors, namely PERK (Protein Kinase RNA-like ER Kinase), IRE1α (Inositol requiring enzyme 1α) and ATF6 (Activating transcription factor 6) (Figure 1) [152,154]. Improving the folding capacity of ER proteins, these stress sensors dissociate themselves from intraluminal-bound GRP78 (glucose regulated protein; an ER resident molecular chaperone also referred to as BiP (Binding immunoglobulin protein)) once it crosses the stress threshold; thereby switching from an inactive to active state, which leads to the stimulation of the UPR^ER^ response [155]. In this stimulatory event, the activation of ATF6 occurs on its cleavage, while activation of the remaining two factors occurs by self-transphosphorylation [11,154,156,157,158].

#### 5.1.1. PERK Lineage of UPR^ER^ Pathway

PERK—a crucial transducer of UPR, is composed of a cytosolic and kinase domain. From attachment to GRP78 in uninduced cells, its activation is marked by homo-dimerization that subsequently progresses to trans-autophosphorylation of its cytoplasmic component, which ultimately leads to the inhibition of eukaryotic cytoplasmic initiation factor 2A (eIF2A) [159,160,161,162]. Phosphorylation of the eIF2A drops the activity of eIF2A levels (eIF2A); thereby provisionally admitting association of Met-tRNA_i_^Met^ to 40S ribosome and as such the architect’s ternary complex of the translational apparatus [163,164]. Although inhibition of the eIF2A restricts the translation process in the majority of proteins in ER [160,162], it fails in doing so for GRP78 and transcription factor ATF4 [165,166]. ATF4 plays an important role in regulating the expression of genes involved in apoptosis, antioxidant response, and amino acid metabolism [167,168,169]. ATF4 upregulates the expression of C/EBP homologous protein (CHOP), whose translocation to the nucleus causes an increase in the BiP level and a simultaneous decrease in the expression of Bcl2 (full form). A decrease in the Bcl2 level enables release of cytochrome C and apoptosis inducing factor (AIF), which triggers fast forward progression to the mitochondrial apoptotic pathway [165,170,171,172]. Additionally, phosphorylated PERK induces nuclear translocation of Nrf2 (an antioxidant cytoplasmic protein associated with the BTB-domain containing protein Keap1); thereby leading to upregulation of the defensive antioxidant response. Nrf2 was found inducing the expression of detoxifying enzymes under in vivo conditions [173,174], besides activating the PERK-eIF2α axis of the NF-κB pathway [175,176].

#### 5.1.2. IRE1 Lineage of UPR^ER^ Pathway

IRE1 is a transmembrane protein with Ser/Thr activity that exists in two forms (IRE1α; ubiquitously expressed in different tissues, and IRE1β; confined to lungs and intestines) among mammals [177,178,179]. Increase in the ER stress results in dissociation of IRE1α from GRP78, which proceeds with the excision of transcription factor X-box binding protein 1 (XBP1); thereby causing a shift in the reading frame, which subsequently leads to an increase in the expression of the stable and active form of XBP1s [180,181]. All this proceeds with the transactivation of genes involved in ER expansion, maintenance and ER-associated degradation (ERAD) [180,182,183,184]. IRE1α-enabled degradation of misfolded glycoproteins is mediated by a change in the expression of Derlin-2 in association with ER degradation enhancing α-mannosidase–like protein (EDEM) and EDEM2 [185,186,187]. Interaction of IRE1α with TNF receptor-associated factor 2 (TRAF2) leads to the activation of JUN N-terminal kinase (JNK) and apoptosis signal-regulating kinase 1 (ASK1) [185,186,187]. JNK modulates the apoptotic process via, regulation in nuclear factor kappa-light-chain-enhancer of activated B cells, BIM and BCL-2 functioning [188,189,190], while ASK1 together with JNK inhibitory kinase (JIK), and ASK1-interacting protein-1 (AIP1) activates the IRE1α-JNK cell death signaling pathway [187,191,192]. Additionally, phosphorylation of JNK causes its translocation across the mitochondrial membrane where it elevates the proapoptotic protein, BaD (Bcl-2-associated death promoter) and Bax (Bcl-2-associated X protein) levels that progresses with serious damage to the mitochondrial membrane [193].

#### 5.1.3. ATF6 Lineage of UPR^ER^ Pathway

ATF6 is a transcriptional factor with CREB/ATF bZIP (basic leucine zipper) domain [194]. Prevalent as an inactive precursor in the ER membrane, its activation by S1P and S2P occurs after translocation to Golgi; whereby it is released as p50ATF-6 (a functional isomer of 50 Kd) [195]. Acting as a third mediator of UPR^ER^ response, its senses ER perturbations on exposure to stressful conditions through its cytosolic C-terminal region. The cytosolic N-terminal cleaved product, p50ATF6 of the full length ATF6 (p90ATF) translocates to the nucleus and binds to ER stress-response elements (ERSE-1) and ATF/cAMP response elements (CRE); thereby acting as a transcription factor associated with the upregulation of genes involved in normal ER functioning, such as GRP78, PDI (Protein Disulfide Isomerase), XBP1, and CHOP [152,196].

### 5.2. ER Stress Induced Apoptosis

In the event of insufficient UPR^ER^ response mediated by stress sensors (PERK, IRE1α and ATF6), the cells progress to the apoptosis (intrinsic as well as the extrinsic pathways) [157,197,198,199]. Major players involved in the apoptotic event include; (i) PERK/eIF2α-dependent induction of the pro-apoptotic transcriptional factor CHOP; (ii) IRE1-mediated activation of TRAF2 stimulating ASK1 and JNK cascade, and (iii) Bax/Bcl2-regulated Ca^2+^ release from the ER. CHOP/GADD153 (growth arrest/DNA damage) is recognized as one of the most important mediators of ER stress-induced apoptosis. In the intrinsic lineage of apoptotic pathway, a signaling platform comprising of BAX or BAK (BCL2 family members) promotes mitochondrial outer membrane permeabilization (MOMP) by assembling themselves on the mitochondrion; thereby triggering apoptosis by promoting a release of various apoptogenic factors as well as cytochrome C [200]. An extrinsic lineage of the apoptotic pathway commences with extracellular stimuli on the transmembrane receptors, the prominent being the TNF-receptor gene superfamily [201,202]. Reports suggest activation of JNK by exogenous stimuli during ER stress [186]. Activation of JNK transduces the signal along the TRAF2 bound to IRE1 (at cytoplasmic component), which invariably promotes activation of caspase-12 [186]. On the other hand, activation of ASK1 was found stimulating p38, MAPK and JNK [187].

### 5.3. ER Stress Induced Autophagy

Autophagy (lysosome-mediated bulk degradation pathway) emerged as a protective mechanism for the elimination of damaged organelles, protein aggregates, and worn-out proteins. To accommodate a high protein load, ER components proliferate, which proceeds with ER-associated degradation (ERAD) of misfolded or unfolded proteins. The ERAD mechanism of protein degradation includes (1) ubiquitin-proteasome-dependent ERAD (autophagy pathway targeting soluble form of misfolded protein) and (2) autophagy lysosome dependent ERAD (autophagy pathway targeting misfolded protein of both soluble and insoluble nature) [203]. Similar to UPR^ER^, autophagy induces cell death under stress conditions. Induction of autophagy proceeds through the release of Ca^2+^ via, inositol 1, 4, 5-trisphosphate receptor channel (IP3R), which results in the phosphorylation of CaMKKβ (Ca^2+^-calmodulin-dependent kinase β) and activation of AMPK (AMP-activated kinase); indirectly associated with inactivation of the ULK1 (autophagy-activating kinase 1) complex through the inhibition of mTOR (Mechanistic target of rapamycin) [204]. The release of Ca^2+^ also results in the activation of DAPK (death-associated protein kinase) which phosphorylates Beclin1 and Bcl2 (B-cell lymphoma-2), which ultimately leads to induction of autophagy via activation of JNK1 [204]. Additionally, XBP1 (spliced) enhance the formation of the microtubule associated protein1 light chain 3 (LC_3_-I/II), which results in decreasing the activity of Fork head box O_1_ (FoxO_1_); thereby triggering Beclin-1. Another arm of UPR activated PERK induced autophagy via expression of Autophagy Protein, ATG_16_L, DNA-damage inducible transcript 3(DDIT_3_), ATG_12_, by ATF4 transcription factor. Similarly, CHOP activates tribbles-related protein 3 (TRIB3), which suppresses the activity of Akt/mTOR pathway and induces autophagy. The ATF6α arm of UPR induces the autophagy through inhibition of phosphorylation in mTOR and Akt pathway.

The extensive exploitation of ER module by cancerous cells turns them immortal and thereby prevents them from undergoing the apoptotic event. There are several findings that demonstrate the prominent role of UPR in cancer cells progression. For instance, in normal cells, GRP78 (an ER-resident molecular chaperone) binds to and inactivates the ER stress sensor, however, in BC or other malignancies, GRP78 is found to be overexpressed [198,205]. The overexpression of GRP78 is attributed for cell survival as peptides specifically targets GRP78 suppressed tumor growth [206].

## 6. FDA Approved Nanotechnological Formulations for BC

Nanotechnology is a technology employed in improving the diagnosis and therapeutics of a disease. Focused on the delivery of the substances in the nanoscale range, the structures developed possess incredible potential to encapsulate a wide range of therapeutic moieties such as protein-based drugs, peptides, nucleic acids, etc. With improved solubility and stability in the biological environment, their release in a controlled manner helps in the maintenance of drug concentration in systemic circulation; thereby enhancing the stability of the drug in circulation and its distribution to the target tissues. Efficient delivery of drugs to target tissues helps in reducing the toxicity of drug cargoes to non-target organs. Nanotechnology based drug formulation approved by the FDA include: Trastuzumab (ADC, antibody-drug conjugate targeting HER2 in HER2 positive BC), Abraxane (nab-paclitaxel, albumin bound nanoparticle bound to paclitaxel), Onivyde (irinotecan liposome injection), and more recently Onpattro (patisiran; first RNAi drug) [38,207,208,209]. These nanomedicines have demonstrated increased bioavailability, enhanced stability, active tumor targeting, and high drug loading, being successfully brought to the market.

## 7. Conclusions and Future Perspectives

Erroneous efforts to develop methods of delivering therapeutic cargo with improved efficacy and reduced toxicity have led to the production of biomaterials capable of delivering therapeutics in a target-specific manner. Despite significant advances, there is still a problem of premature release of the therapeutic payload, which decreases the treatment outcome and often becomes a cause in inducing systemic toxicity. As safety remains the principal concern for efficient translation of nanoparticle-based therapeutics from bench to bedside, exploration of efficient formulations of nanomaterial that prolong the stability of therapeutics in systemic circulation for optimum delivery at the target site with minimum potential off-target effects of the therapeutic cargo loaded on to nanoparticles. Anticipating improvement in delivering therapeutic cargo, camouflaging the nanoparticles with a surface covering the biocompatible material or cell membrane conjugated to a target-specific ligand for improving the stability, and as such selective delivery of the therapeutics to tumors in the response of stimuli, would avoid a non-specific or pre-mature release of the therapeutics. Taking into consideration the surface composition of the nanoparticles correlating with the stability of therapeutics in systemic circulation for the selective and efficient release of the therapeutics, engineering of nanoparticles bearing these properties would remove the impeding obstacles of nanoparticles for use in clinical applications towards improving patient outcome.

## Figures and Tables

**Figure 1 biomedicines-09-00635-f001:**
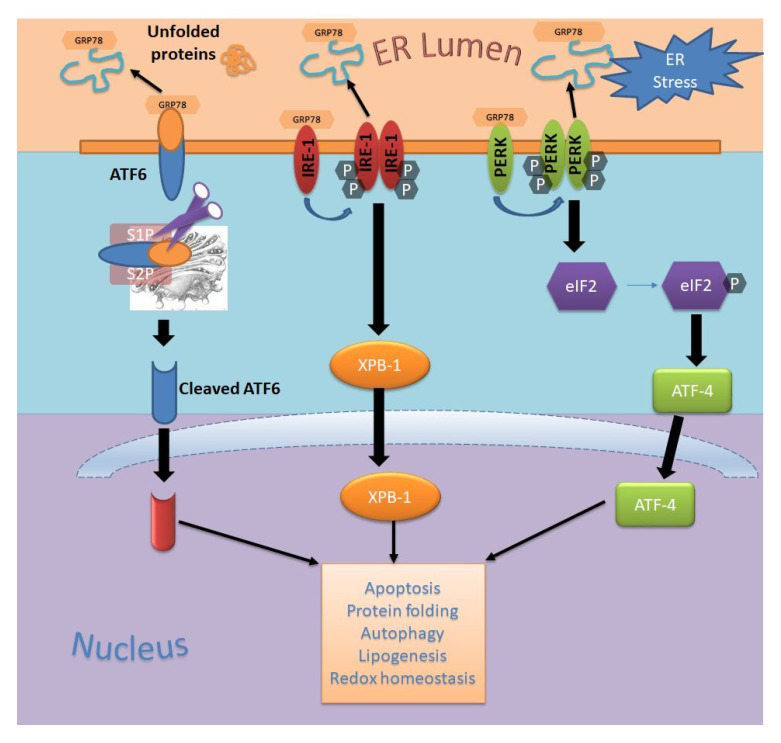
Endoplasmic reticulum stress mediated cell signaling pathway.

**Table 1 biomedicines-09-00635-t001:** Information of representative inhibitors currently undergoing clinical trials for TNBC.

Inhibitor Name	Clinical Implications	Phase	NCT Number
**Epidermal Growth Factor (EGF) receptor inhibitors**			
Afatinib	Pan-HER tyrosine kinase inhibitor	Phase II	NCT02511847
Dasatinib	Pan-Src tyrosine kinase inhibitor	Phase II	NCT02720185
**Vascular Endothelial Growth Factor (VEGF) receptor inhibitors**			
Sunitinib	Inhibition of proliferation, invasion, and apoptosis resistance in TNBC cell lines	Phase I/II	NCT00887575
Apatinib	Tyrosine kinase inhibitor targetting VEGFR2	Phase II	NCT01176669
**Poly ADP-Ribose Polymerase (PARP) inhibitor**			
Iniparib	Non-competitive PARP inhibitor	Phase II	NCT01045304
Veliparib	PARP1/2 inhibitor	Phase II	NCT01306032
**Fibroblast Growth Factor (FGF) receptor inhibitor**			
Dovitinib	FGFR, VEGFR and PDGFR inhibitor	Phase II	NCT01528345
Lucitanib	FGFR1/2/3 and VEGFR1/2/3 inhibitor	Phase II	NCT02202746
**Androgen Receptor (AR) inhibitors**			
Bicalutamide	Androgen antagonist	Phase III	NCT03055312
Seviteronel	Cytochrome P450c17a inhibitor	Phase I/II	NCT02580448

Source: http://ClinicalTrials.gov (Accessed date, 26 May 2021).

**Table 2 biomedicines-09-00635-t002:** Activation and modulation of ER stress, unfolded protein response, and autophagy in treating breast and other cancer cells/models.

Formulations of NPs	Model-Type	Remarks	Reference
Dual peptide-decorated melanin-like NPs	In vitro: breast cancer Hela, and MDA-MB-231 cells In vivo: BALB/c nude mice	Beclin 1 promoted autophagy, but activated minimal in vitro cytotoxicity, sensitized cancer cells to photothermal therapy thereby enhancing significant cell killing, RGD- mediated tumor targeting increased the tumor targeting for well-regulated inhibition of tumor growth at a mild temperature	[117]
Fe_3_O_4_ NPs	In vitro: breast cancer In vivo: mice	Causes ER stress, which includes autophagy and apoptosis Extensive accumulation of autophagosome in the kidney and spleen	[118]
PAV-AuNPs	In vitro: triple-negative breast cancer cells) MDA-MB-231 cells	Activation of autophagy	[119]
RQDs	In vitro: endometrium carcinoma JEC cells	Induction of apoptosis and necrosis via, ER stress	[120]
AANTs-TG-3MA	In vitro: Breast cancer cell MDA-MB-231-TXSA, HFF, and THP-1 cells	AANTs showed good biocompatibility, 3MA, at a non-toxic dose, reduced the autophagy-level, and improved the cell killing effect of AANTs-TG, TG and 3MA in combination enhanced the ER stress signaling	[121]
AgNP (2 and 15 nm)	In vitro: human MCF-7 and T-47D breast cancer cells	Induced upregulation of the transcription factors ATF-4 and GADD153/CHOP and induction of apoptosis.	[122,123]
AgNPs (75 nm)	In vitro: human MCF-7 and T-47D breast cancer cells In vivo: TNBC tumor xenografts mice.	MCF-7 and T-47D are more sensitive to MCF-10A cells and Induced ER stress in TNBC cells Effective at non-toxic doses for reducing the growth	[124]
AgNPs-EPSae (AgNPs-specific polysaccharide)	In vitro: human breast (SKBR3 and 8701-BC)	ER stress, oxidative stress and mitochondrial impairment triggering cell death trough apoptosis and/or autophagy activation.	[125]
AgNPs-EPSae (AgNPs-specific polysaccharide)	In vitro: human breast (SKBR3 and 8701-BC)	ER stress, oxidative stress, and mitochondrial impairment triggering cell death trough apoptosis and/or autophagy activation.	[125]
PLGA NPs loaded with LY294002	In vitro: lungs cancer cells H157, H460, H1650 In vivo: Xenograft animal model (female athymic nude mice)	Accumulation and pronounced induction of ER stress, activation of JNK and prominent in vivo antitumor effect	[126]
FTIC labeled-PEI-PLGA-PTX-MNPs	In vitro: human brain glioblastoma U251 cells	NPs effectively endocytosed by targeted U251 cells, induction of cell apoptosis, and autophagy	[127]
PEG-PE micelles	In vitro: A549 lungs cancer cells	Accumulation and induction of ER stress via disturbing ER lipid homeostasis, high expression of CHOP, and proapoptotic Bax/Bak in cancer cells	[128]
PEGylated nanogels containing AuNPs	In vitro: SCCVII (murine) or A549 (human lung) cells	Enhance cell radiosensitivity, activation of ER stress, JNK activation, and induction of apoptosis,	[129]
LP-SeNPs	In vitro: human liver carcinoma HepG2 cells	Inhibition of autophagy and activation of mitochondria pathway for the induction of apoptosis	[130]

## Data Availability

Not applicable.

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
