# Peer review of "Molecular Perspective of Nanoparticle Mediated Therapeutic Targeting in Breast Cancer: An Odyssey of Endoplasmic Reticulum Unfolded Protein Response (UPR^ER^) and Beyond"

_biomedicines, 2021, doi:10.3390/biomedicines9060635_

Round 1
Reviewer 1 Report
- Manuscript have to be checked for typo. Examples : line 131 50 nm - 10 µm or line 204 "formulation in in nude"
- I suggest to the authors a list of abbreviations used as some are not defined in the text. Example : PTX
- Section 3.4. on Inorganic Nanoparticle. The authors discuss the mechanisms by which cancer cell growth is affected by metallic nano-objects. The effects of metal nanoparticles on cell biology is a field that gained in interest in recent years. Yet, they refer to a 2014 paper (ref 81). I suggest expanding this section by recommending reading the following papers: Penninckx et al, Cancers, 2020; Turnbull et al, ACS Nano 2019; Schuemann et al, Phys. Med. Biol 2020.
- Table 1a & b are really interesting but the authors have to accentuate the discussion around the modulation of the UPR response by nanoparticles.
Author Response
Authors thank reviewer for the concerns raised that significantly improved the content of the manuscript
- the whole manuscript has been corrected for typos and errors.
- All abbreviations has been written in full at its first occurrence in the text.
- Section 3.4 has been revised strictly as per the suggestion and taken into consideration the papers advised by the reviewer.
- The information is part of the 2nd paper submitted in continuation to this which is at present under consideration. A separate table has been added.
Reviewer 2 Report
Reviewer Blind Comments to Author
This manuscript is a topical review on using nanomedicine (nanoparticle-based) in breast cancer treatment. This work is well organized and comprehensive. The topic is quite novel and timely. I only have some minor comments for further improvement:
- Abstracts: When mentioning FDA, the authors should clarify if it is the FDA of the USA.
- It is good to report how the authors acquired the references in this work. That is, what search engine and database they used.
- The authors should explain the reason why they need to prepare such topical review. Is it the suitable time to do so or they should in fact wait for more works published first. What is the aim to prepare this work, and how this work can help the readers?
- Introduction: For mentioning the background nanomaterials in cancer therapy and drug delivery, some recent important references such as Siddique et al (Nanomaterials, 2020, 10, 1700) and Siddique et al (App Sci, 2020, 10(11), 3824) should be added.
- 3.4: When discussing inorganic NPs, some recent works such as Moore et al (Nano Ex. 2021;2:022001) should be added.
- 3.4: L245: When discussing DNA damage due to ionizing radiation, important works such as Chun et al (AIMS Bioeng. 2016;3:352) should be added.
- 4: Similar to Sec. 3, it is good to provide a table to summarize all the mechanistic insights in breast cancer treatments with related references.
- The authors are encouraged to add more figures and tables in the manuscript. Currently there is only one figure in the manuscript.
Author Response
Authors thank reviewer for the concerns raised that significantly improved the content of the manuscript
- The FDA is having reference to US, so modified accordingly in the abstract section.
- A separate heading of methodology has been added that covers the concern and suggestion raised by the reviewer.
- The authors have put together information covered from 1991-present. It is designed so that persons working in the field could get all necessary information at one stop. It will help the scientific fraternity by making it a start point for further necessary work to be undertaken
- Necessary information in the introductory section along with the references suggested by the reviewer has been added at specific points in the introduction.
- Section 3.4 has been revised in the light of reviewer suggestions.
- A separate table has been added as per the suggestion.
Round 2
Reviewer 1 Report
Thanks to the authors for taking my comments into account
Reviewer 2 Report
I am satisfied with the corrections and modifications that made by the authors regarding my concerns.